



# Stratospheric tropical warming event and its impact on the polar and tropical troposphere

Kunihiko Kodera[1], Nawo Eguchi[2], Hitoshi Mukougawa[3], Tomoe Nasuno[4], Toshihiko Hirooka[5]

[1]Institute for Space-Earth Environmental Research, Nagoya University, Nagoya, Japan
5  [2]Research Institute for Applied Mechanics, Kyushu University, Kasuga, Japan
[3]Disaster Prevention Research Institute, Kyoto University, Uji, Japan
[4]Research Institute for Global Change, Japan Agency for Marine-Earth Science and Technology, Yokohama, Japan
[5]Department of Earth and Planetary Sciences, Kyushu University, Fukuoka, Japan

*Correspondence to*: Kunihiko Kodera (kodera @isee.nagoya-u.ac.jp)

10  **Abstract.** Stratosphere–troposphere coupling is investigated in relation to middle atmospheric subtropical jet (MASTJ) variations in boreal winter. An exceptional strengthening of the MASTJ occurred in association with a sudden equatorward shift of the stratospheric polar night jet (PNJ) in early December 2011. This abrupt transformation of the MASTJ and PNJ had no apparent relation to the upward propagation of planetary waves from the troposphere. The impact of this stratospheric event penetrated into the troposphere in two regions: in the north polar region and the tropics. Due to the strong MASTJ 15  planetary waves at higher latitudes were deflected and trapped in the north polar region. Trapping of the planetary waves resulted in amplification of zonal wavenumber 1 component, which appeared in the troposphere as the development of a trough over the Atlantic sector and a ridge over the Eurasian sector. A strong MASTJ also suppressed the equatorward propagation of planetary waves, which resulted in weaker tropical stratospheric upwelling and produced anomalous warming in the tropical stratosphere. In the tropical tropopause layer (TTL), however, sublimation of ice clouds kept the temperature 20  change minor. In the troposphere, an abrupt termination of a Madden Julian Oscillation (MJO) event occurred following the static stability increase in the TTL. This termination suggests that the stratospheric event affected the convective episode in the troposphere.

## 1 Introduction

Stratosphere–troposphere dynamical coupling is an important factor for tropospheric weather and climate. The influence of 25  downward penetration of zonal winds from the polar stratosphere, such as the annular modes (Baldwin and Dunkerton, 1999; Thompson and Wallace, 2001) or the polar night jet (PNJ) oscillation (PJO) (Kuroda and Kodera, 1999, 2004; Hitchcock et al., 2013), has been well documented. More recently, the connection between tropospheric weather and changes in planetary wave structure in the polar region, due to reflection or downward propagation in the polar region, has also been reported (Perlwitz and Harnik, 2003; Shaw and Perlwitz, 2013; Kodera et al., 2008, 2013, 2016a). Although stratosphere–troposphere 30  coupling in the tropical region is more controversial, a possible connection has been proposed based on the modulation of deep convective activity by the stratospheric quasi-biennial oscillation (QBO) (Collimore et al., 2003; Liess and Geller,





2012; Yoo and Son, 2016) and sudden stratospheric warming (SSW) (Kodera, 2006; Eguchi and Kodera, 2010; Kodera et al., 2015; Eguchi et al., 2016).

External forcings in the stratosphere can affect the troposphere through the above-mentioned stratosphere–troposphere coupling processes. For instance, a stratospheric ozone depression can produce surface pressure change through modulation of the Southern Hemisphere (SH) annular mode (SAM) (Thompson and Solomon, 2003; Marshall et al., 2004; Polvani et al., 2011). The 11-year solar cycle also affects the surface through modulation of the annular mode or PJO in the Northern Hemisphere (NH) winter stratosphere. However, the solar signal does not appear in the SAM (Lu et al., 2011; Kodera et al., 2016b). Instead, middle atmosphere subtropical jet (MASTJ), which is usually maximised around 30° latitude and 0.5 hPa, extends downward into the stratosphere in the SH (Kodera et al., 2016b). It has been suggested that the MASTJ, related to the solar cycle, modulates stratospheric mean meridional circulation, which further influences convective activity in the tropical troposphere (Kodera, 2004; Kodera and Shibata, 2006). On the other hand, it has been pointed out that the MASTJ may be important for the formation of a kind of dynamical instability in the extratropical mesosphere (Iida et al., 2014).

The role of MASTJ in stratosphere–troposphere coupling is poorly understood. The MASTJ in the SH stratopause region strengthens until nearly winter solstice, whereas in the NH the MASTJ starts to decay earlier, before the winter solstice (Kodera and Kuroda, 2002). When the upward propagation of planetary waves increases in mid-winter in the NH, mesospheric MASTJ shifts poleward and weakens (Dunkerton, 2000). Therefore, downward extension of the MASTJ in the NH winter circulation has not attracted attention. In early December 2011, however, an exceptionally rapid downward extension of the MASTJ from the lower mesosphere to the bottom of the stratosphere occurred. Such a sudden change enables us to investigate the evolution of downward penetration. In the present study, we investigate this particular event that produced global tropospheric impacts in the north polar region as well as in the tropics.

The remainder of the paper is organized as follows. The data are described in Section 2, and the results of an analysis of stratosphere–troposphere coupling during November–December 2011 are presented in Section 3. The stratospheric processes producing a strong MASTJ in the stratosphere are first presented in subsection 3.1. Stratosphere–troposphere coupling is realized by two different processes. In the NH extratropics, coupling occurs through change in planetary wave propagation, while in the tropics, tropospheric deep convection responds to change in the static stability of the tropical tropopause layer (TTL) induced by the stratospheric mean meridional circulation. These processes are presented in subsections 3.2.1 and 3.2.2, respectively. For intraseasonal variability in the tropics, the Madden–Julian Oscillation (MJO) (Madden and Julian, 1972) is a typical phenomenon. We briefly argue plausible influence of the rapid downward extension of the MASTJ on MJO activity.





## 2 Data

In this study we use meteorological reanalysis datasets produced by the Japan Meteorological Agency, JRA−55 (available from the web page http://jra.kishou.go.jp/JRA-55/index_en.html) (Kobayashi et al., 2007). Unless otherwise specified, anomalies are defined as departures from the 37-year climatological mean (1979–2015). The data have a horizontal

resolution of 1.25° by 1.25° and 37 vertical levels, of which 10 levels are above 100 hPa with a top at 1 hPa. Cloud fraction data in the TTL are derived from the Cloud Layer Product (Level–2, ver. 3.01) from the Cloud-Aerosol LIdar with Orthogonal Polarization (CALIOP) aboard the CALIPSO satellite (Winker et al., 2007). We also use ice water content data measured by the Earth Observing System/Microwave Limb sounder (EOS/MLS) (Level–2, ver.4.2x) (Livesey et al., 2015) at 146 hPa. Daily data on a 2.5° (latitude) by 2.5° (longitude) grid are first derived from the orbital data. Outgoing longwave

radiation (OLR) data provided by NOAA (e.g. Arkin and Ardanuy, 1989) are used to analyse convective activity in the tropics. The Tropical Rainfall Measuring Mission (TRMM) daily integrated precipitation data (TRMM 3B42 v7) are used to study surface precipitation (Huffman et al., 2007).

## 3 Results

### 3.1 Stratospheric event

Two types of westerly jet form in the middle atmosphere during winter: the MASTJ in the subtropics and the PNJ around the polar region. The MASTJ is forced primarily by solar ultraviolet (UV) heating in the tropics, whereas the PNJ is associated with longwave cooling in the polar region. However, planetary waves also interact with these westerly jets and modulate the MASTJ and PNJ in a complicated way. The climatology of the evolution of the NH zonal-mean zonal wind is presented in Fig. 1a from 1 October to 1 February. The PNJ in the middle stratosphere at 10 hPa increases until January, whereas MASTJ

at 1 hPa reaches its maximum around 20 November. After November, the MASTJ decreases and shifts poleward.

To show a particular characteristic of the interannual variation of the MASTJ and PNJ, Fig. 1b presents a scatter diagram of zonal-mean zonal wind in the subtropical stratopause region (25°N−35°N, 1 hPa) and polar stratosphere (60°N−70°N, 10 hPa) during a period of high velocity of the MASTJ (16−30 November), for the period 1979–2015. There is a negative

correlation, such that stronger MASTJ is associated with weaker PNJ and vice versa. In the case of the 2011 winter (red circle, Fig. 1b), however, the MASTJ and PNJ are both strong. This exceptional situation in at the end of November (Fig. 1c) soon dissolves during the following period (1–15 December) by transforming into a more usual structure of a stronger MASTJ accompanied by relatively weak PNJ (Fig. 1d). To illustrate the circulation change around 4 December, differences between the two 10-day means of zonal-mean zonal wind and temperature are shown in Fig. 2. The change in zonal wind

appears as a deep seesaw between the MASTJ and the PNJ extending across the whole stratosphere from the stratopause to the tropopause. An important change in the temperature field occurs as warming in the tropics accompanied by cooling in



mid-latitudes of the NH, consistent with the strengthening of the MASTJ. The tropical warming extends farther southward into the summer hemisphere, up to 45°S. In addition, a narrow warming region around the North Pole occurs in association with a decrease of the PNJ.

The evolution of zonal-mean zonal wind during November–December 2011 is illustrated in Fig. 3c. The MASTJ at 1 hPa (colour shading) shows a continuous increase from mid-November to December, while the PNJ at 5 hPa (contours) does not increase in November and largely decreases after 4 December. Vertical sections of zonal winds around 60°N−70°N and 35°N−45°N are shown in Fig. 3d and 3e, respectively. Zonal winds in the subtropics largely increase from 4 December, while the PNJ decreases thereafter. The weakening of the PNJ is delayed at lower altitudes (Fig. 3d). This delay suggests that

the interaction between the MASTJ and PNJ starts at upper levels and gradually extends downward. The increase of MASTJ in the middle and lower stratosphere involves an equatorward shift of the PNJ (Fig. 3c).

Simultaneous changes are also found in the stratospheric planetary wave field. Figure 3b shows zonal asymmetric components of geopotential height averaged over 60°–70°N at 10 hPa. The amplitude of the planetary waves is small during

November, but a ridge and trough develop from early December around 180° and 0° longitudes, respectively, which makes a wavenumber 1 feature more conspicuous. This modification of wave structure at 10 hPa suggests a change in propagation property in the stratosphere. The amplification of the stratospheric wave in early December is not related to an increase in the eddy heat flux at 100 hPa averaged over 45°N–75°N, a measure of the vertical propagation of planetary waves from the troposphere (Fig. 3a). This indicates that the transition of the circulation at the beginning of December has a middle

atmospheric rather than tropospheric origin. The eddy heat flux at 100 hPa largely increases after 17 December, leading to a minor warming in the upper stratosphere, and both the MASTJ and PNJ weaken towards the end of December (Fig. 3c). The impact of this event is not limited to the extratropics. Changes in the zonal-mean zonal wind are associated with modification of the meridional circulation. Figure 3f shows zonal-mean anomalous pressure vertical velocity in the tropical region (20°S–20°N) of the middle stratosphere. Anomalous downwelling develops concurrently with strengthening of the MASTJ from 4

to 17 December (Fig. 3f). This change manifests as a warming in the tropical stratosphere down to the tropopause (Fig. 3g). Note that the tropical upper troposphere shows a slight cooling during this period.

### 3.2 Impact on the troposphere

### 3.2.1 Extra-tropics

The bottom panels of Fig. 4 show 5-day mean zonal-mean zonal wind and the Eliassen–Palm (E–P) flux (e.g., Andrews et

al., 1987) from the end of November to early December, corresponding to a period of rapid transformation of the stratospheric westerly jet. The time tendency of the zonal-mean zonal wind in the upper stratosphere is displayed in the top

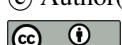

panels. The time tendency of zonal mean wind ($U$) on day $n$ is calculated from the difference between the 3-day mean before and after day $n$ as $\Delta U_n = (U_{n+3} + U_{n+2} + U_{n+1} - U_{n-3} - U_{n-2} - U_{n-1})/12$.

In spite of the increasing upward component of E–P flux in the lower stratosphere around the core of the PNJ from the end
of November to the beginning of December, the subtropical westerly jet in the upper stratosphere-stratopause region continued to growth, although the PNJ in the stratopause region decreased. This suggests rather passive role of the upward propagation of planetary wave on the evolution westerly winds in the subtropical stratopause. Waves in the stratosphere converge more at higher latitudes, since the equatorward propagation tends to be hindered by stronger MASTJ. This leads to a large deceleration of the PNJ and a suppression of the upward propagation of planetary waves in the polar region around 8
December. Accordingly, waves in the polar region are trapped in the stratosphere.

Change in the planetary wave propagation in the polar region can also be seen in the evolution of the vertical wave structure. Figure 5a shows height–longitude sections of the zonally asymmetric component of geopotential height averaged over 60°N–70°N. When the PNJ is strong on 28 November, waves propagate upward from the troposphere guided by the PNJ
(Matsuno, 1970). The planetary waves propagate upward as a wave packet composed of multiple zonal wavenumber components (Hayashi, 1981). Upward propagation can be seen as westward tilted ridge and trough lines with increasing altitude over the Eurasian sector (Fig. 5a). In the upper stratosphere, waves are deflected equatorward due to the stronger PNJ. Therefore, the wave amplitude in the upper stratospheric polar region is small at the end of November. On 3 December, the wave amplitude increases in the upper stratosphere. The westward tilt of trough and ridge lines is still conspicuous over
the Eurasian sector, indicating the persistent upward propagation in the sector; however, the trough line tilts eastward over the Atlantic sector. Thus, the wave in polar region is trapped and deflected in the upper stratosphere and propagates downward in the Atlantic sector. When the PNJ further weakens and the MASTJ becomes stronger on 8 December, the upward propagation of the planetary waves is largely suppressed. The standing wave feature (i.e., little phase tilt in the vertical) indicates that the amplification of the wavenumber 1 component occurs due to interference between the upward and
downward propagating waves. The impact on the troposphere can be seen in the 500 hPa geopotential height (Fig. 5b). Because the wave activity is low, meandering is weak on 28 November. When wavenumber 2 components are trapped in the middle stratosphere, a ridge develops over the North Pacific on 3 December. Finally, amplification of the wavenumber 1 component results in the development of a trough over the Atlantic sector and a ridge over the Eurasian sector, forming a blocking over the Eurasian continent.

**3.2.3 Tropics**

To investigate the downward penetration of stratospheric variation in the equatorial SH, time–height sections (averaged over the Equator to 20°S) of zonal-mean temperature tendency, and normalized 3-day mean vertical pressure velocity are shown in Fig. 6a and 6b, respectively. An increase in temperature tendency in early December is apparent at levels higher than 100



hPa (Fig. 6a). The upper tropospheric temperature tendency is opposite to that in the lower stratosphere. The variation of the temperature tendency corresponds well to that of the standardized pressure vertical velocity within the stratosphere (Fig. 6b). However, unlike the negative temperature tendency, the positive anomaly of vertical pressure velocity extends farther into the troposphere after around 4 December, which coincides with a period of decreased ice cloud fraction in the TTL (Fig. 6c).

The changes in vertical velocity in the lower stratosphere lead the temperature change (Fig. 3g) by ~3 days, starting and ending around 1 and 14 December (vertical lines in Fig. 6).

When considering the lower tropospheric temperature tendency, the adiabatic heating produced by anomalous downwelling would be compensated by the sublimation of ice cloud, and therefore the temperature does not increase in the upper

troposphere. The decrease in convective activity in the troposphere over the equatorial SH is also indicated by positive anomalies in outgoing longwave radiation (OLR) (Fig. 6d). As the precipitation also decreases (Fig. 6e), both clouds in the upper troposphere and deep convective clouds decrease during this period. These changes are opposite to that observed during the cooling phase in the tropical stratosphere related to SSW events (Eguchi et al., 2015; Kodera et al., 2015). In fact, opposite changes in the tropics are seen after 15 December 2011 that are associated with the occurrence of a minor SSW

event (Fig. 6): the tropical stratospheric temperature decreases (Fig. 6a), ice clouds become abundant in the TTL (Fig. 6c), and precipitation increases at the surface (Fig. 6e).

Changes in the TTL during the tropical stratosphere warming event are depicted in Fig. 7. Consecutive 7-day mean height–latitude sections are calculated for (a) anomalous temperature, (b) cirrus cloud frequency, and (c) vertical velocity during the

periods of (i) 24−30 November, (ii) 1−7 December, and (iii) 8−14 December. For a clearer view of the vertical extent of the tropospheric upwelling in the TTL, vertical pressure velocity is converted to vertical velocity in Fig. 7c. To illustrate the evolution of the vertical temperature gradient in the TTL, anomalous temperature in Fig. 7a is shown as the difference between each pressure level and 200 hPa. Descent of warm anomalies from the lower stratosphere to the TTL is clearly seen through these periods. Cirrus clouds largely decrease in the equatorial TTL, and a weakening of upwelling (corresponding to

a strengthening of anomalous downwelling in Fig. 6b) is seen in the troposphere during period (ii). Suppression of upwelling continues in the SH during period (iii), while some recovery is seen in the NH. This difference creates some meridional asymmetry, which is also evident in the cloud field.

To show the horizontal structure, anomalous OLR is presented for the same three periods in Fig. 8. An intense convective

centre is located over the Indian Ocean at the end of November ((i) in Fig. 8). This centre of action moves eastward over the Maritime Continent and then weakens (ii). During period (iii) when upwelling in the equatorial SH troposphere is suppressed, positive anomalies in OLR expand over the Indian Ocean. On the other hand, negative anomalies are distributed more zonally in the tropical NH.





The current analysis period is included in the field experiment "CINDY/DYNAMO" campaign to collect in-situ atmospheric and oceanic data to study MJO over the equatorial Indian Ocean (Yoneyama et al., 2013). One of the characteristics of the tropical circulation during this boreal autumn–winter is that the MJO was particularly active (e.g., Nasuno 2013). An MJO event in 2011 started on 17 September and ended on 8 December according to Gottschalck et al. (2013). The phase diagram of the multivariate MJO index of Wheeler and Hendon (2004), from 21 November to 31 December 2011, is presented in Fig. 9. The period of tropical warming in the lower stratosphere from 4 to 17 December in Fig. 1 is indicated by red solid lines. The periods before and after are indicated by dark blue solid lines and black dashed lines, respectively. Eastward propagation of the MJO is apparent during November 2011. The MJO suddenly weakens in the Maritime continent region around 8 December. This disruption of the MJO follows the enhanced equatorial anomalous downwelling in the troposphere (Fig. 7b).

The multivariate index is a combination of three different variables (OLR, zonal winds at 850 and 200 hPa). Figures 10a and 10b show specific humidity and velocity potential at 925 hPa, respectively, averaged around the equator (10°S–10°N) during November to December 2011. The eastward propagation of the convergent area (positive anomaly of the velocity potential) following an increase of anomalous water vapour is observed in December and November in the lower troposphere. The MJO is characterized by a convergence zone in the lower level and a divergence zone above. At 200 hPa (Fig. 10c), the divergent (negative anomaly) area propagates eastward in November, similar to the 925 hPa level. To facilitate the comparison with the lower level, negative values are show by warm colours at 200 hPa. The divergent area at 200 hPa does not propagate eastward after 8 December and disappears over the eastern Pacific, in spite of the conditions being favourable to maintain the convective activity, such as an increase in convergence at 925 hPa (Fig. 10b) and an increase in moisture (Fig. 10a). Anomalous static stability in the TTL is presented in Fig. 10d as the difference between the temperature at 100 and 200 hPa. An increase in stability in the TTL (after 4 December) is concurrent with the termination of the MJO.

## 4 Discussion and concluding remarks

In the present study, we investigated circulation changes related to the formation of a strong MASTJ in the stratosphere in November–December 2011. The event started from the mesosphere, penetrated to the troposphere in the polar region, as well as in the tropics, and involved diverse amplifying processes. During November 2011, the MASTJ and PNJ were both strong around the stratopause region. Usually, the MASTJ shifts poleward and becomes weaker when the planetary wave activity increases with the seasonal march (Fig. 1a). However, in the case of December 2011, the stratospheric PNJ abruptly shifted equatorward before the poleward shift of the MASTJ (Fig. 3c), which was the unique event. This sudden change in the MASTJ and PNJ was not closely related to the tropospheric forcing change. Since meridional zonal wind shear in the tropics has a large impact on the propagation of planetary waves (Yoden and Ishioka, 1993), a strong MASTJ and weak PNJ tend to deflect the equatorward propagation of planetary waves in the upper stratosphere. This deflection would further accelerate the MASTJ and decelerate the PNJ due to decreased and increased convergence of E–P flux in the tropics and polar region,





respectively. Concurrent temperature change occurred in the NH stratosphere (Fig. 2b), with warming in the tropics and cooling in mid-latitudes of the entire stratosphere, consistent with the thermal wind balance. These circulation changes in the stratosphere had a significant impact on the troposphere.

The exceptional event in early winter 2011-2012 may be attributed to a strong MASTJ and PNJ in November. The strong PNJ may have resulted from weak wave forcing in the troposphere (Fig. 3a). However, the strong MASTJ coexisted with an enhanced PNJ in November 2011, which is an unusual situation (Fig. 1b). As the top level of the JRA-55 dataset is 1 hPa, it is difficult to undertake further investigation. A preliminary study using Microwave Limb Sounder data indicates that the strong MASTJ first formed in the mesosphere and extended downward into the stratosphere. Therefore, analysis of the
mesosphere is crucial for investigating the origin of this event, which will be done in a separate study.

During the downward penetration of the middle atmospheric circulation change, stratospheric planetary waves deflected by the stronger MASTJ were trapped in the polar region and their amplitude increased over time (Fig. 3b). Some parts of wave packets propagating upward from the Eurasian sector were trapped in the lower stratosphere and troposphere, which led to
amplification of a ridge over the North Pacific (Fig. 5). Further trapping of the wave resulted in amplification of the zonal wavenumber 1 component in the polar region. This trapping was also associated with the development in the troposphere of a trough over the Atlantic sector and a ridge over the Eurasian sector, leading to the formation of a blocking there. Such a change in the wave structure is similar to that observed following planetary wave reflection events (Shaw and Perlwitz, 2013; Kodera et al., 2008, 2013), although the pattern was somewhat shifted eastward in the present case. In the usual case,
an initial change in the zonal mean zonal wind field in the stratosphere is created by a stronger upward propagation of planetary waves from the troposphere (Kodera et al., 2016a). In the present case, reflection occurred without the preceding enhanced upward propagation of planetary waves from the troposphere, but was attributed to the deflection of waves in the upper stratosphere by MASTJ.

The development of the MASTJ associated with the deflection of planetary waves (Fig. 5a) resulted in suppressed upwelling in the tropical stratosphere (Fig. 3f). According to the downward control principle (Haynes et al., 1991), momentum forcing induces meridional circulation below the level of the zone where the forcing is applied. In addition, the meridional extent of the induced circulation becomes wider than that of the forcing when a transient response is considered (Holton et al., 1995). In this respect, the wave forcing change related to the MASTJ should have a stronger impact on the tropics than that of the
PNJ. If we focus on the tropics alone, the present strong MASTJ event was opposite to the SSW that induces enhanced tropical upwelling (Eguchi et al., 2015; Kodera et al., 2015). In addition, the temperature change in the troposphere was negligible during the present event, but the vertical velocity change penetrated farther into the TTL and troposphere due to interaction with clouds. Thus, it is expected that adiabatic heating due to the anomalous downwelling would be balanced by the induced diabatic cooling by cloud evaporation in the TTL (Figs. 6 and 7).





The impact of stratospheric circulation on the MJO has been reported in connection to the QBO (Kuma, 1990; Yoo and Son, 2016). The present results suggest that the change in TTL stability due to anomalous stratospheric downwelling had an impact on the sudden termination of the MJO event in early December 2011 (Fig. 10), opposite to the situation under an
SSW event (Eguchi et al., 2015).

**Acknowledgements**

This work was supported in part by JSPS Grants-in-Aid for Scientific Research (S) 2422401, (C) 25340010, (B) 26287115. EOS/MLS and CALIOP data were from Atmospheric Science Data Center (ASDC) at NASA. Analyses of TRMM data used in this paper were produced with the Giovanni online data system, developed and maintained by the NASA GES DISC.

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





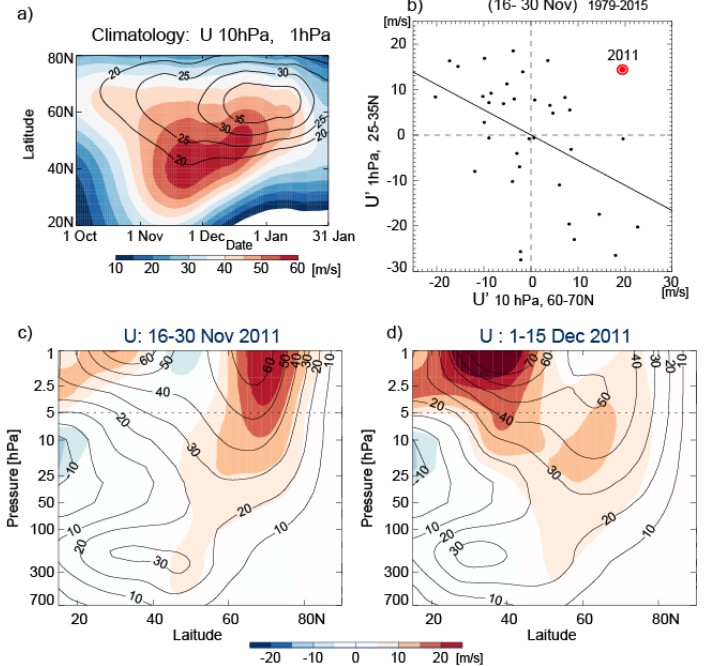

**Figure 1: Latitude–time section of the climatological zonal-mean zonal wind at 1 hPa (colour shading) and 10 hPa (contours). b)**
**Scatter diagram of the zonal-mean zonal wind averaged from 16 to 30 November over 25°N–35°N at 1 hPa (ordinate) and 60°N–**
**70°N at 10 hPa (abscissa) for the period 1979–2015. The solid line indicates a regression line. c) Height–latitude section of the**
**zonal-mean zonal wind averaged from 16 to 30 November 2011. Contours and colour shading indicates the climatology and**
**anomalies, respectively. d) Same as c) but averaged from 1 to 15 December 2011. Vertical lines indicate 4 and 17 December,**
**periods of tropical warming in the lower stratosphere.**





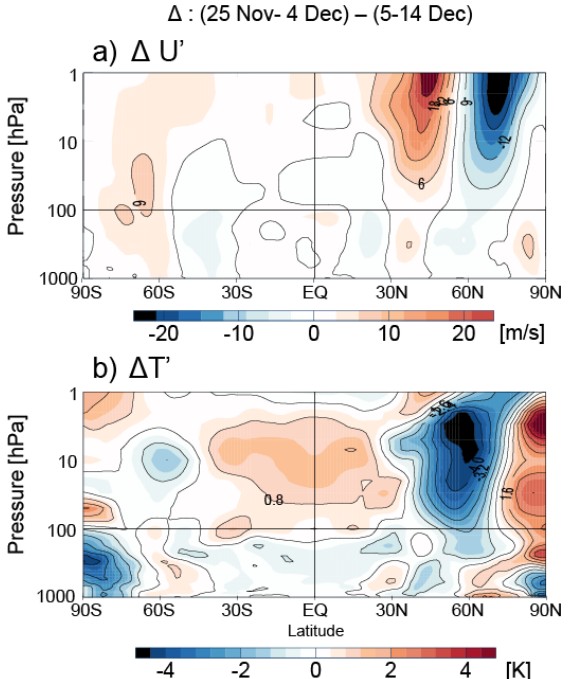

5  **Figure 2: Difference between the mean values from 25 November to 4 December 2011 and from 5 to 14 December 2011 for (a) anomalous zonal-mean zonal wind and (b) anomalous zonal-mean temperature.**





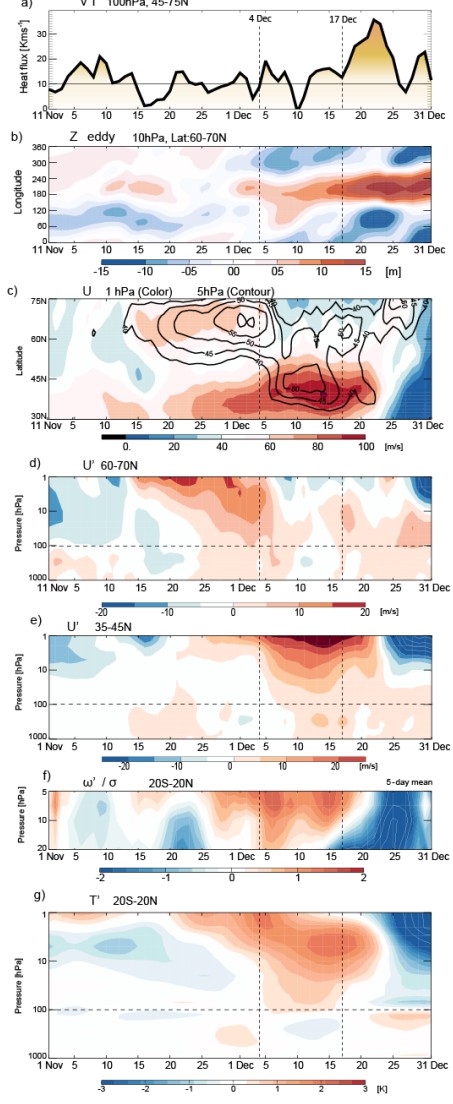

**Figure 3: a)** Eddy heat flux at 100 hPa averaged over 45°N–75°N. **b)** Zonally asymmetric component of the geopotential height averaged over 60°N–70°N at 10 hPa. **c)** Latitude–time section of zonal-mean zonal wind at 1 hPa (colour shading) and 5 hPa (contours). **d)** Height–time section of anomalous zonal-mean zonal winds averaged over 60°N–70°N. **e)** Same as d) but for zonal wind averaged over 35°N–45°N. **f)** Anomalous vertical pressure velocity averaged over 20°S–20°N. 5-day running mean has been applied for (f). **g)** Same as (f) but for temperature. Analysis period is from 1 November to 31 December 2011, and the time mean field during the period is further subtracted from anomalies in (f) and (g).



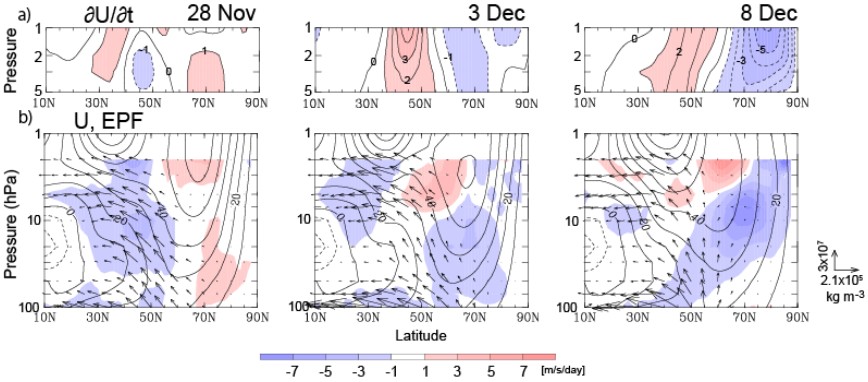

**Figure 4: a) Zonal-mean zonal wind tendency. [ms⁻¹/day] b) Five-day-averaged zonal-mean zonal winds (contours) and E–P flux (arrows). E–P flux divergences are also shown using colour shading. The dates from left to right (28 November, 3 December, and 8 December) are the middle day of the 5-day mean.**

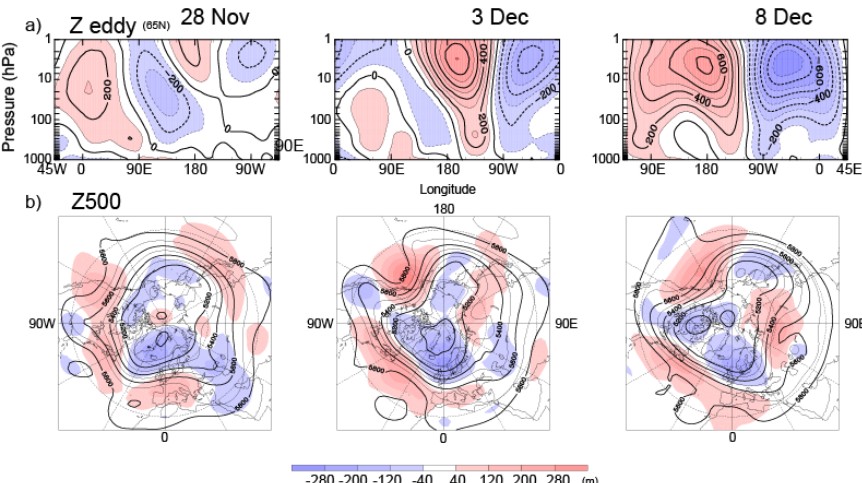

**Figure 5: a) Latitude–height section of the 5-day-mean zonally asymmetric component of geopotential height [m] averaged over 60°N–70°N. The origin of the longitude shifts eastward with time: 45°W, 0°, and 45°E from left to right. b) The 5-day-mean 500 hPa geopotential height (contours) and deviation from the climatology (colour shading). The analysis period is the same as in Fig. 4.**





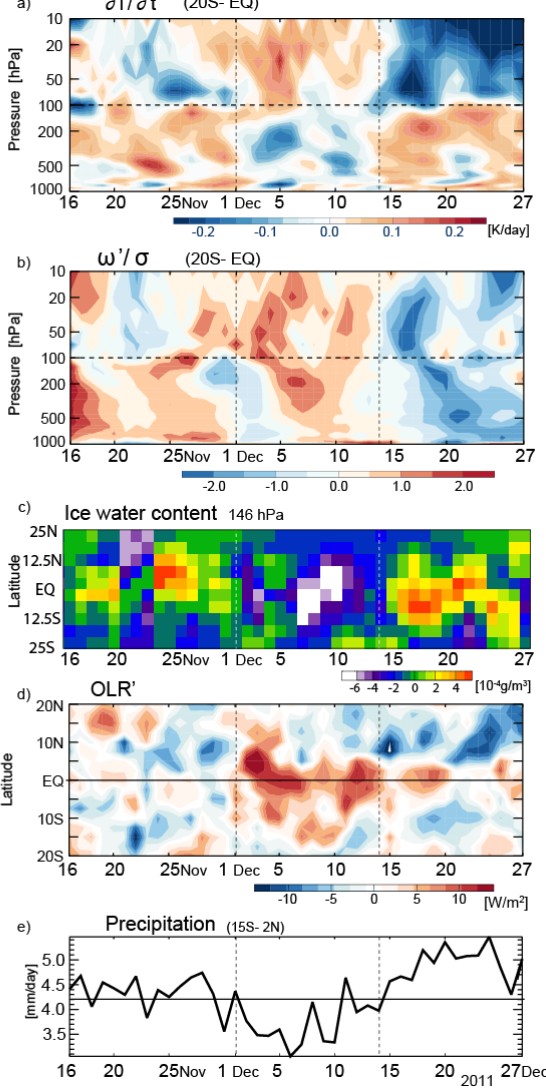

Figure 6: a) Height–time section of zonal mean temperature tendency averaged from 20°S to the Equator. b) Same as a) but for a 3-day running mean of the anomalous vertical pressure velocity normalized by the daily standard deviation. (c) Latitude–time section of ice water content at 146 hPa, of which linear tendency has been subtracted. (d) Same as (c) but for anomalous OLR. (e) Surface precipitation averaged from 15°S to 2°N, as estimated from TRMM.





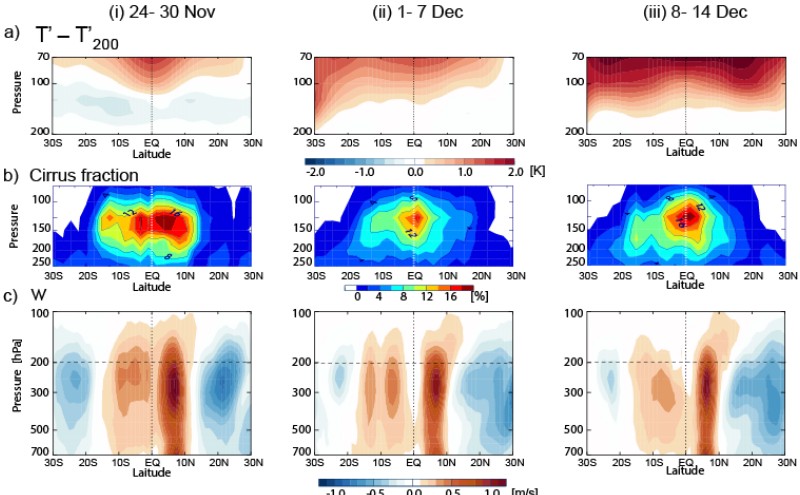

**Figure 7: Consecutive 7-day-mean height–latitude sections: (i) 24–30 November, (ii) 1–7 December, and (iii) 8–14 December. a) Anomalous temperature difference between each pressure level and 200 hPa. b) Cirrus cloud frequency, and c) vertical velocity.**

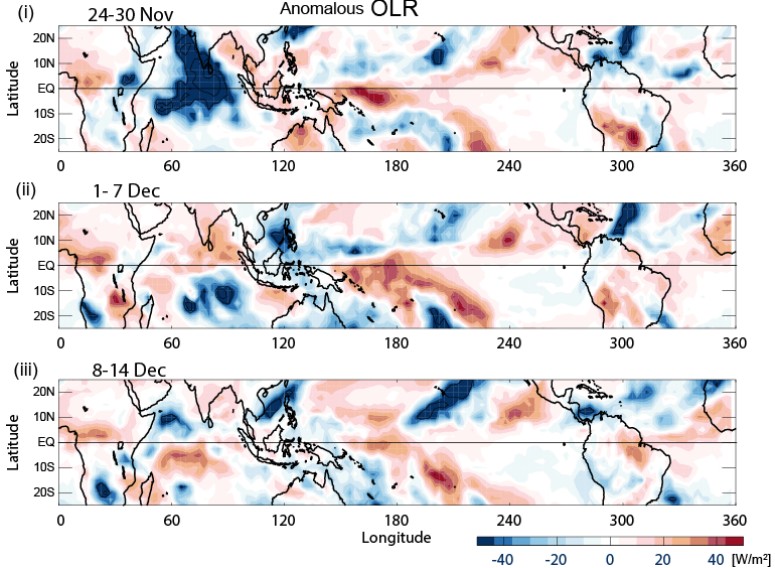

**Figure 8: Same as Fig. 7, but for latitude–longitude sections of OLR anomalies.**


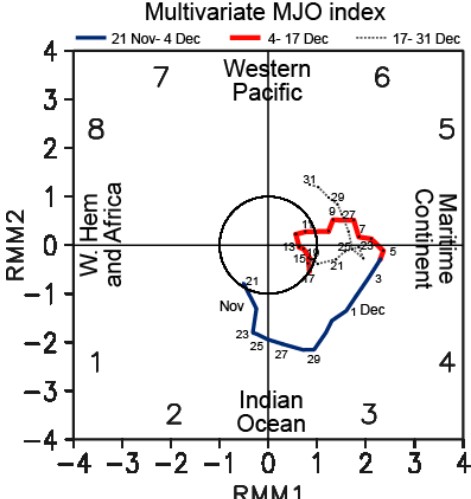

Figure 9: Phase diagram of the multivariate MJO index from 21 November to 31 December 2011.

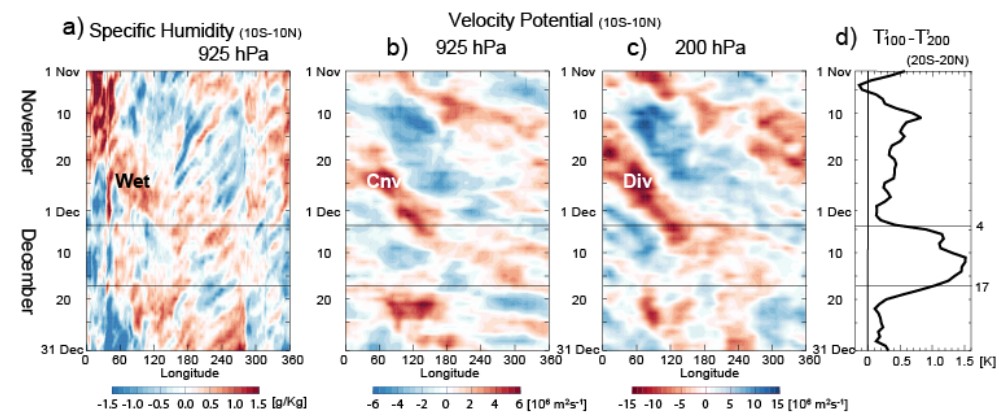

Figure 10: Time–longitude section around the equator (10°S–10°N) during November to December 2011 for (a) anomalous specific humidity, (b) anomalous velocity potential at 925 hPa, (c) anomalous velocity potential at 200 hPa, and (d) difference in anomalous temperature between 100 and 200 hPa. Time mean values during the analysed period are subtracted from climatological
10   anomalies. 'Wet', 'Cnv', and 'Div' indicate wet, convergent, and divergent zones, respectively. Horizontal lines mark 4 and 17 December, similar to Fig. 1.