# Peer review of "Stratospheric tropical warming event and its impact on the polar and tropical troposphere"

_Atmospheric Chemistry and Physics, 2016_

## Referee Comment (RC1) · Anonymous Referee #1 · 3 Nov 2016

This manuscript presents an exceptional stratosphere-troposphere coupling event associated with a strengthening of the middle atmospheric subtropical jet and a rapid change of the polar night jet. This study is very interesting not only for the description of a very particular dynamics event occurring in the middle atmosphere during the Northern winter but also for the understanding of its impact on the tropical tropospheric convection. It merits to be published in Atmospheric Chemistry and Physics after taking into account the minor remarks given below.

1) Section 3.2.1 page 4 lines 29-31 and page 5 lines 1-10 and Figure 4: I would expect to have a better correlation between the EP flux divergence (Fig4 bottom) and the acceleration of the zonal flow (Fig 4 top). For instance on 28 November the two quantities are anticorrelated at 30°N and correlated at 50 to 70°N. Please could you explain why it is not the case.

[Figure]

2) The term "anomalous" is used for the zonal-mean wind and zonal-mean temperature in Fig. 2, for vertical pressure velocity in Fig. 3 and at several places in the text. Please explain what it means. Is it a deviation from a climatological mean?

Page 6, lines 25-27: It is indicated that some recovery of the upwelling is seen in the NH form period (ii) to period (iii). When I look at the Figure 7c, this is not obvious. The upwelling between 5 and 10°N seems to be about of the same amplitude. It may be a problem of representation.

---

## Referee Comment (RC2) · Anonymous Referee #2 · 3 Nov 2016

Summary: The manuscript describes an event in Nov/Dec 2011, where an anomalous strengthening occurred in the middle atmosphere subtropical jet (MASTJ), a band of strong winds at the subtropical stratopause. Exceptionally, the stratospheric polar night jet (PNJ), which tends to be anticorrelated in strength with the MASTJ, shifted equatorward in an event that the authors argue was due to internal stratospheric dynamics, as opposed to tropospheric forcing. Only later during the event did tropospheric wave propagation set on and further weaken the PNJ. This stratospheric anomaly had an effect on the tropospheric circulation, both by changing the refractive properties of the stratosphere for tropospheric wave propagation in the extratropics, but also through an impact on the TTL, changing tropical convection and leading to an abrupt change in the MJO.

General assessment: The authors develp a clean case study of a phenomenon that is

otherwise not yet well represented in the literature, but which confirms earlier studies on stratosphere - troposphere coupling on wave reflection and the impact of the tropical lower stratosphere on tropical convection. The study is well documented, but could use a consolidation of figures for clarity. Overall, the case study is well written and supported by the recent literature, mainly from the authors themselves. This study is worth publishing after minor revisions to the text and figures.

Detailed assessment: page 1, line 25: "downward penetration of zonal winds" is misleading, it should rather be called a downward propagation of anomalies, cite e.g. Plumb & Semeniuk (2003) page 2, line 4: "external forcings in the stratosphere": external to the troposphere? page 2, line 5: "ozone depletion"? page 2, line 8: please add a reference page 2, lines 8 - 10: It would be helpful to explain the MASTJ itself before looking at its impact. One option would be to move part of the first paragraph in section 3.1 (page 3, lines 15 - 20) to the introduction. page 2, line 16: nearly? page 3, line 26: remove "in" page 4, line 20: "largely" -> "strongly" page 5, line 2: where does the 12 come from? page 5, line 6: growth -> grow page 5, line 8: it looks more like divergence in the polar regions, please be more clear in the description Figure 1: a) missing in caption Figure 2a): there seems to be a similar structure in the SH as in the NH, but much weaker

---

## Author Comment (AC1) · 3 Dec 2016

Many thanks for reading our manuscript and your comments.

Please find our answers to your comments.

1) Section 3.2.1 page 4 lines 29-31 and page 5 lines 1-10 and Figure 4:

I would expect to have a better correlation between the EP flux divergence (Fig4 bottom) and the acceleration of the zonal flow (Fig 4 top). For instance on 28 November the two quantities are anticorrelated at 30N and correlated at 50 to 70N. Please could you explain why it is not the case.

> Acceleration of the zonal wind is determined by combination of EP flux divergence and Coriolis force. The anticorrelation of EP flux divergence with acceleration of the zonal wind suggests that planetary wave forcing is not the major factor to accelerate zonal wind. The zonal wind may be accelerated by EP flux divergence of unresolved smaller-scale wave or Coriolis force due to meridional circulation driven by diabatic heating.

> The following sentence has been added in p.5 line 7-9
> Acceleration of subtropical zonal wind in the upper stratosphere on 28 November could be resulted from unresolved wave (gravity waves) forcing, and/or increased mean meridional circulation due to diabatic heating.

2) The term "anomalous" is used for the zonal-mean wind and zonal-mean temperature in Fig. 2, for vertical pressure velocity in Fig. 3 and at several places in the text. Please explain what it means. Is it a deviation from a climatological mean?

> Yes. According to the comment, the following sentence has been added in p.4 line 24
>    "Here, the term "anomalous" means deviation from a climatological mean (1979-2015)."

Page 6, lines 25-27:   It is indicated that some recovery of the upwelling is seen in the NH form period (ii) to period (iii). When I look at the Figure 7c, this is not obvious. The upwelling between 5 and 10N seems to be about of the same amplitude. It may be a problem of m of representation.

> Color shading has been slightly modified so as the difference becomes more visible.

---

## Author Comment (AC2) · 3 Dec 2016

Many thanks for reading our manuscript and your comments.
Please find our answers to your comments.

General assessment: The authors develop a clean case study of a phenomenon that is otherwise not yet well represented in the literature, but which confirms earlier studies on stratosphere - troposphere coupling on wave reflection and the impact of the tropical lower stratosphere on tropical convection. The study is well documented, but could use a consolidation of figures for clarity. Overall, the case study is well written and supported by the recent literature, mainly from the authors themselves. This study is worth publishing after minor revisions to the text and figure.

Detailed assessment:

page 1, line 25: "downward penetration of zonal winds" is misleading, it should rather be called a downward propagation of anomalies, cite e.g. Plumb & Semeniuk (2003).

Some peoples consider that the term "propagation" is not appropriate, because the process in the stratosphere is not exactly the same as that in the troposphere.

In the case of Plumb and Semeniuk (2003), they prescribed periodic variation of tropospheric wave source and argued that the downward "migration" of zonal winds does not necessarily reflect downward influence from the stratosphere. Therefore, they addressed a somewhat different problem from the present study.

page 2, line 4: "external forcings in the stratosphere": external to the troposphere?

External to the atmosphere, such as the solar forcing as described in the following sentences.

page 2, line 5: "ozone depletion"?

Ozone depletion due to ozone depleting substances (ODS)

page 2, line 8: please add a reference

Reference added: Kodera et al. (2016b)

page 2, lines 8 - 10: It would be helpful to explain the MASTJ itself before looking at its impact. One option would be to move part of the first paragraph in section 3.1 (page 3, lines 15 - 20) to the introduction.

According to the comment, the paragraph has been moved to the introduction.

page 2, line 16: nearly?

This means that it is not exactly until winter solstice.

page 3,line 26: remove "in"

Removed

page 4, line 20: "largely" -> "strongly"

Modified

page 5, line 2: where does the 12 come from?

12= 3 (days) x 4 (days interval between (n+2) and (n-2))

For easier understanding, the equation has been rewritten as follows.

$$\Delta U_n = [(U_{n+3} + U_{n+2} + U_{n+1})/3 - (U_{n-3} - U_{n-2} - U_{n-1})/3]/4.$$

page 5, line 6: growth -> grow

Corrected

page 5, line 8: it looks more like divergence in the polar regions, please be more clear in the description previous

"Waves in the stratosphere converge more at higher latitudes, ...." This sentence describes the change from the end of November to the beginning of December, as mentioned in the preceding two sentences. Convergence of the EP-flux is evident at higher latitudes on 8 December as illustrated below. According to the comment, the word "December" is added in the phrase as follows.

"...converge more at higher latitudes in December."

[Figure]

Figure 1: a) missing in caption

Added.

Figure 2a): there seems to be a similar structure in the SH as in the NH, but much weaker.

The structure of both hemispheres is compared in the following figure. Meridional structure of zona winds looks rather different.